# Mucins and CFTR: Their Close Relationship

**DOI:** 10.3390/ijms231810232

**Published:** 2022-09-06

**Authors:** Kenichi Okuda, Kendall M. Shaffer, Camille Ehre

**Affiliations:** 1Marsico Lung Institute, School of Medicine, University of North Carolina at Chapel Hill, Chapel Hill, NC 27599, USA; 2Department of Pediatrics, School of Medicine, University of North Carolina at Chapel Hill, Chapel Hill, NC 27599, USA

**Keywords:** mucus, mucins, cystic fibrosis (CF), CFTR, single-cell transcriptomics, airway clearance, polymeric network, viscoelasticity

## Abstract

Mucociliary clearance is a critical defense mechanism for the lungs governed by regionally coordinated epithelial cellular activities, including mucin secretion, cilia beating, and transepithelial ion transport. Cystic fibrosis (CF), an autosomal genetic disorder caused by the dysfunction of the cystic fibrosis transmembrane conductance regulator (CFTR) channel, is characterized by failed mucociliary clearance due to abnormal mucus biophysical properties. In recent years, with the development of highly effective modulator therapies, the quality of life of a significant number of people living with CF has greatly improved; however, further understanding the cellular biology relevant to CFTR and airway mucus biochemical interactions are necessary to develop novel therapies aimed at restoring *CFTR* gene expression in the lungs. In this article, we discuss recent advances of transcriptome analysis at single-cell levels that revealed a heretofore unanticipated close relationship between secretory MUC5AC and MUC5B mucins and CFTR in the lungs. In addition, we review recent findings on airway mucus biochemical and biophysical properties, focusing on how mucin secretion and CFTR-mediated ion transport are integrated to maintain airway mucus homeostasis in health and how CFTR dysfunction and restoration of function affect mucus properties.

## 1. Introduction

Cystic fibrosis (CF) is an autosomal genetic disorder characterized by the dysfunction of the cystic fibrosis transmembrane conductance regulator (CFTR) protein, an ion channel transporting Cl^−^ and HCO_3_^−^ that is expressed in various tissues, notably in mucus-producing organs [1]. One of the primary drivers of CF pathogenesis is the accumulation of a thick, adherent mucus, particularly in the lungs, gut, and pancreatic ducts [2]. Dysfunction of CFTR causes fluid hyperabsorption and low HCO_3_^−^ concentrations, which can affect mucus network organization in numerous ways that will be discussed in this review [3,4]. The aberrant properties of CF mucus in the lungs affect mucociliary clearance (MCC) and result in airway muco-obstruction, chronic inflammation, bacterial infection, decline in lung function, and, eventually, respiratory failure.

CF was first described in 1938 by Dr. Dorothy Andersen as “cystic fibrosis of the pancreas” after she observed histological sections of the pancreas of children who died from malnutrition [5]. In 1945, recognizing that CF affected more than the pancreas, Dr. Sydney Farber referred to the disease as “mucoviscidosis” due to the abnormally thick mucus that individuals with the disease produced [6]. Once considered a fatal disease of childhood, the advancement of therapeutics and treatment options for CF patients over recent decades increased life expectancy to nearly 50 years. In the early days, commonly used therapies focused on addressing symptoms and reversing airway obstruction via physiotherapy, bronchodilators, osmotic agents, and mucolytics/recombinant human DNase (rhDNase) [2,7,8]. While effective at slowing the progression of lung disease, these therapeutic approaches failed to address the root cause of the disease, CFTR dysfunction.

CFTR mutations are grouped in six different classes according to their impact on protein synthesis (Class I causing absence of CFTR protein), folding (Class II causing trafficking defects), or function (Class III–VI causing defects in channel gating, quantity, and/or stability). In the past decade, CFTR modulators, which consist of small molecules acting systemically to restore CFTR function, have gradually changed the way physicians care for CF patients and, more importantly, have significantly improved the quality of life of patients eligible to take these medications. In 2012, ivacaftor (VX-770), a CFTR potentiator, was the first modulator therapy to be approved to restore CFTR function with gating mutations, such as G551D [9,10,11]. In addition to improved airway clearance and pulmonary function (e.g., FEV_1_), patients undergoing ivacaftor treatment experienced increased BMI, decreased sweat chloride concentration, fewer hospitalizations, and reduced incidence of *Pseudomonas aeruginosa* infection. Although this compound was later approved for additional mild mutations (e.g., P67L, R117H), only a small fraction (<10%) of people living with CF could benefit from taking this medication [12]. 

In the following years, research focused on correcting the function of the most common mutation, F508Del, as roughly 85% of the CF population carries at least one copy of the class II mutation. Two additional modulator therapies were developed combining ivacaftor with a CFTR corrector compound, lumacaftor (VX-809) or tezacaftor (VX-445) [13]. While both therapies, lumacaftor/ivacaftor (LUM/IVA or Orkambi) and tezacaftor/ivacaftor (TEZ/IVA or Symdeko), were shown to improve overall lung function in patients homozygous for the F508Del mutation, efficacy remained modest compared to ivacaftor in patients with gating mutations [14,15]. In 2019, a triple-combination modulator drug, elexacaftor/tezacaftor/ivacaftor (ETI or Trikafta), combining VX-660, VX-445, and VX-770, was approved for the treatment of patients with at least one copy of F508Del mutation. The addition of a second CFTR corrector, elexacaftor, to TEZ/IVA resulted in significantly increased drug efficacy, producing a 15% increase in FEV_1_ and a 63% decrease in pulmonary exacerbations [16,17,18]. 

In the era of highly effective CFTR modulators, the quality of life of many patients living with CF has improved considerably. However, a fraction of patients remains unable to benefit from these drugs due to a different class of mutations (e.g., non-sense) or adverse reactions to CFTR modulators. To address the root cause of CF and provide treatment for all patients, scientific efforts are currently focusing on the development of *CFTR* gene editing and gene transfer, which rely heavily on novel molecular approaches (e.g., readthrough, mRNA transfer/repair, short nucleotide therapies). Other key elements of gene manipulation are the identification of accurate cell targets and the bypassing of physical obstacles, such as innate defense mechanisms and the thick mucus layer in CF. This review outlines how recent advances of transcriptional analysis have improved our knowledge of CFTR and mucin expression in the lungs, revealing regional and cellular specificity for fluid and mucin secretion by comparing the proximal and distal airways, as well as the superficial epithelia and submucosal glands (SMGs). The close proximity of cell types expressing CFTR and mucins suggests a strong link between mucus production and ion transport. We also highlight the effects of CFTR function on mucin network organization, since reversing aberrant CF mucus properties will be critical to ensure successful DNA/mRNA delivery to the targeted epithelial cells in the lungs. This review offers insights into the direct relationship between CFTR and secretory mucin functions.

## 2. Affected Regions in CF Lung Disease

### 2.1. Role of CFTR in the Lungs

In light of the fact that CFTR is a transmembrane channel transporter for chloride and bicarbonate, CF lung disease pathogenesis reflects abnormal ion transport. Despite general agreement on this notion, controversy remains in identifying the specific links between abnormal ion transport and CF lung disease. In CF, defective ion transport produces: (1) reduction in airway surface liquid (ASL) volume, mucus hyperconcentration, and an early muco-inflammatory state predisposing to bacterial infection [19,20]; and (2) abnormalities in ASL pH leading to defects in airway epithelial host innate defense properties, resulting in persistent bacterial infection [21,22,23]. In addition to pathophysiological consequences related to abnormal ion transport, it is important to understand which compartments of the lung are affected in CF, as these specific regions can be the primary targets for therapeutic strategies. CFTR dysfunction typically causes no change to the alveolar region but critically affects the conducting airways, comprised of two distinct regions: (1) the proximal/large (tracheobronchial) airways that contain submucosal glands (SMGs) and cartilage; and (2) the distal/small (bronchiolar) airways (<2 mm at the diameter) that constitute the major surface area of the airways within the lung [24,25,26].

### 2.2. Structural and Regional Specificity of the Lungs

Compartmentalization of the lung raises two important questions: (1) how do the superficial epithelial or SMG compartments contribute to disease; and (2) which airway region, large or small, is the disease-initiating/vulnerable region or the starting point of CF pathogenesis? Studies aimed at answering these questions have shed light on the structural and regional specificity of the lungs.

The SMGs, restricted to the large airways, are a source of electrolytes, fluid, host defense proteins, and secretory mucins, predominantly MUC5B [27,28,29,30,31]. Compared to the superficial epithelium that produces mucus at baseline and maintains constant directional motion, the SMGs secrete large volumes of mucus following adrenergic or cholinergic stimulation to aid in host defense and galvanize airway clearance. In CF, dysfunctional SMG mucus secretion contributes to impaired MCC in the large airways. Small airways lack SMGs, and the superficial epithelium must clear mucus from the distal regions, a process that relies heavily on the coordinated beating of the cilia. However, mucus hyperconcentration can cause compression of the cilia and subsequent mucostasis [32]. The limited backup mechanism, i.e., lack of SMGs, to clear mucus in small airways predisposes this region to impaired MCC and disease. Studies have postulated that the superficial epithelium of small bronchiolar airways is one of the most severely affected regions in CF [19,20].

### 2.3. Clinical Observations Related to Small Airway Diseases in CF

Evidence for small airways as the central site of CF pathogenesis has been based on pathology [33,34,35,36,37,38,39], pulmonary function [38,39], imaging [40,41,42,43,44], and lower airway sampling studies [19,36,37]. Pathologic studies of CF lungs, including CF children who died early of the disease, have identified small airway mucus plugs as a routine feature of this syndrome [33,34]. Micro-CT studies of CF lungs harvested at the time of transplant revealed that mucus plugs develop after roughly the 6th generation of airways and progressively increase in frequency in the distal bronchiolar regions [45]. Moreover, pulmonary function tests designed to detect regions of airflow obstruction confirmed the presence of small airway obstruction as the first detectable spirometry measure associated with CFTR dysfunction [38,39]. Consistent with this finding, radionuclide MCC studies have demonstrated the predominance of peripheral (small airways) over central (large airways) clearance defects [40,46]. Further supporting these observations, studies of bronchoalveolar lavage fluids performed on CF preschoolers have shown that the incipient mucus material harvested from CF lungs consisted of abundant and irregular/rough MUC5B/MUC5AC mucus flakes, indicating disturbance of mucus homeostasis in the distal airways in CF lung disease (see mucin distribution and effects on mucin network below) [19].

### 2.4. Airflow, Shear Forces, and Mucus Clearance in the Small Airways

Regional observations may explain why CF small airways are so vulnerable to mucus plugging. First, studies describing the relationship between pulmonary airflow and mucus adhesion to cell surfaces suggest that airflow-induced shear forces required to dis-adhere and clear mucus are restricted to the most proximal region (trachea) [47]. In the small airways, airflow rates are ~2 logs lower than in the proximal airways, which correlates with proportionate decrements in shear forces and provides physical reasons for small airways vulnerability in CF [48]. Second, the absence of submucosal gland secretions is likely to contribute to the failure of mucus clearance in the small airway regions. Third, ciliated cells in the small airways are sparse and possess shorter cilia [49,50], limiting the mechanical forces for mucus transport and peripheral MCC [51]. These findings highlight the importance of understanding how each MCC component contributes to increased CF mucus viscoelasticity and/or is associated with the failure of MCC in the small airway regions. In the next chapters, we focus on site-specific expression and function of secretory mucins and CFTR, both of which are major components that regulate airway mucus biophysical properties.

## 3. Regional Distribution of Secretory Mucins in the Conducting Airways

### 3.1. Mucin Concentrations in Health and Disease

MUC5AC and MUC5B, the dominant gel-forming mucins present in the mucus layer lining the airways, play distinct pathophysiological roles in the lungs pertaining to their individual expression patterns and biochemical/biophysical properties [52,53,54,55]. In normal airway secretions, the concentration of MUC5B is reported to be 10 times higher than MUC5AC, with the latter detected in trace amounts in healthy subjects [56,57]. While total mucin concentration increases in a variety of muco-obstructive lung diseases, including CF [20,58], non-CF bronchiectasis [59], COPD [57,60], and asthma [61], the ratio of MUC5AC/MUC5B differs depending on disease phenotype. Compared to healthy subjects, both MUC5AC and MUC5B proteins are elevated in adult CF airway secretions [20,58] and in COPD sputa [57,60], while elevated MUC5AC but not MUC5B is a hallmark of asthmatic sputum [61,62].

### 3.2. Airway Mucins and Their Functions

Different organs produce MUC5AC and MUC5B in the human body; for instance, the stomach secretes MUC5AC, and the female reproductive tract secretes MUC5B, suggesting distinct functional properties. Studies with Muc5b-deficient mice demonstrated that Muc5b was required for airway defense and MCC [63], despite the fact that the large Muc5b SMG reservoir is constrained to the proximal trachea in mice [64]. In contrast, Muc5ac-deficient mice showed no defect in airway defense and/or clearance [63]. Studies performed on human bronchial epithelial (HBE) cell cultures showed that a pathological MUC5AC-rich mucus induced by IL-13 stimulation is more adherent to airway epithelial surfaces compared to the typical MUC5B-rich mucus, which correlates with reduced mucociliary transport (MCT) [62,65]. A recent study analyzed macromolecular mucin assembly using Calu3 cells genetically manipulated to produce either MUC5AC or MUC5B [66]. Electron and atomic force microscopy revealed that MUC5B adopted a linear pattern, while MUC5AC exhibited a high degree of branching. Quartz crystal microbalance (QCM) dissipation analyses indicated that MUC5AC formed a denser, stiffer, and more viscoelastic mucus with a higher order of oligomerization, providing an explanation for the more adhesive properties of MUC5AC as compared to MUC5B. A mouse model overexpressing Muc5ac in the lungs demonstrated protection against influenza infection via the trapping of viruses through the terminal sialic acids cloaking the mucin protein core [67]. In contrast, mice lacking Muc5ac were unable to expulse enteric parasites swiftly [68]. These in vivo and in vitro models demonstrated that MUC5AC and MUC5B have distinct functions and both mucins are required to protect the lungs against inhaled particles and/or pathogens.

### 3.3. MUC5AC and MUC5B Distribution in the Respiratory Tree

In addition to distinct biophysical properties, the unique distribution of MUC5B and MUC5AC in the lungs contributes to their specific roles (i.e., gliding vs trapping). The classic paradigm for the human respiratory tree described MUC5B as a predominant feature of SMG secretion and MUC5AC as a marker of goblet cells that originated from the superficial epithelium after resident cells adopted a mucin-producing phenotype [27,69]. However, data generated from mice indicated that the superficial epithelium secretes Muc5ac along with Muc5b, which was mediated by non-goblet, secretory club cells [70,71]. Recent human studies conducted on healthy subjects confirmed extensive MUC5B mRNA and protein expression from both SMGs and superficial epithelial cells [72,73]. Transcript and protein mapping for MUC5AC and MUC5B from the trachea to the terminal bronchioles in healthy subjects revealed different expression patterns for these secretory mucins, with the most striking difference located in the small conducting airways, while the terminal and respiratory bronchioles remain mucin-free areas [73]. In healthy individuals, cells positive for the club cell secretory protein (CCSP) produce both MUC5AC and MUC5B throughout the large conducting airways. Hence, both mucins are expressed in the proximal airways. At approximately the 10th generation of bronchiolar airways (diameter <2 mm), MUC5AC expression ceases while MUC5B expression remains, leaving MUC5B as the sole secretory mucin produced in the distal airways. Morphometric calculation determined that MUC5B expressed in the surface area of the small airways is slightly greater than MUC5B expressed in the SMGs in the lung, designating the distal airway superficial epithelium as the major source of MUC5B in the lungs. While both MUC5B and MUC5AC protein levels were elevated in CF airway secretions, the expression of *MUC5B* transcripts predominantly increased in the small airway epithelia, suggesting the importance of MUC5B overproduction in this region in CF small airway pathogenesis [74].

## 4. Cell Types Expressing CFTR in Human Conducting Airways

### 4.1. Early Findings on Cell Types Expressing CFTR

Determining the cell types responsible for CFTR expression and function is fundamentally important for the development of effective therapies based on CFTR gene transfer/repair. Targeting the appropriate cells that normally express CFTR in the lungs and understanding how these cells use innate defense mechanisms are critical steps for the development of novel molecular approaches aimed at restoring CFTR gene expression. Over the last decades, significant controversy has surrounded the cell types expressing CFTR in human airways. Early studies based on immunohistochemistry indicated the localization of CFTR within non-ciliated cells in the superficial epithelium, including CK14-positive basal cells in the large airways [75] and CCSP-positive cells in the small airways [76]. Conversely, a separate study showed CFTR protein at the apical plasma membrane of ciliated cells in the superficial epithelium [77]. Several possible limitations to these morphometric studies may explain the discrepancy, including sensitivity and specificity of antibodies targeting CFTR [78], differences in airway regions examined, and lack of systematic quantitation for CFTR signals. However, multiple original studies identified a rare cell type called “hot cells” with intense CFTR signals in the submucosal gland ducts and the superficial epithelium of human large airways [75,77,79]. Although reported over a decade ago, these cells had not been studied until very recently (see next paragraph).

### 4.2. New Insights with Single-Cell Transcriptional Profiling

The latest technological advancements have allowed for transcriptome analyses at the single-cell level, resulting in the identification of new cell types in multiple organs within a wide range of species [80,81,82]. Single-cell transcriptional profiling studies, focusing on human and mice large airways, identified *CFTR*-rich epithelial cells, called pulmonary ionocytes [83,84], as the potential “hot cells” identified in the early morphometric studies. A compelling study showed a positive association between the number of pulmonary ionocytes and the activity of CFTR in HBE cell cultures, despite ionocyte numbers accounting for a small fraction (<1–2%) of the total cells [84]. In contrast, a different study demonstrated no loss of CFTR-mediated Cl^−^ secretory function in HBE cells in which ionocytes were genetically depleted by CRISPR/Cas9 technology targeting *FOXI1*, a key transcription factor specific to ionocyte lineage [72]. Thus far, it remains unclear whether pulmonary ionocytes contribute significantly to Cl^−^ and HCO_3_^−^ secretion in the lungs. Data from mouse trachea revealed that 54.4% of *Cftr* transcripts were concentrated in ionocytes, despite this cell type comprising <1% of the total cells, suggesting that ionocytes and at least another cell type play a role in CFTR expression and function in the murine airways [83]. More recently, comprehensive single-cell transcriptome analyses performed on human lungs identified secretory club cells as the most common cell type expressing *CFTR* transcripts in the large and small airway regions [85]. Intraregional characterization of *CFTR* and *FOXI1* transcripts along the proximal–distal axis of the lungs revealed discrepancies between robust *CFTR* expression and trivial numbers of ionocytes in the distal airway regions, suggesting that secretory club cells may convert to the dominant cell type expressing functional CFTR in the small airways (see Figure 1). Another study using single-cell transcriptional profiling revealed that the secretory cell is the most common cell type expressing *CFTR* in the non-CF superficial epithelium, which remains true in the CF trachea, despite significant dynamic transcriptome alterations in CF [86]. In CF tracheas, specific changes in transcriptional profiles of secretory cells included overactivated secretory functions, exhausted metabolic profiles, and elevated endoplasmic reticulum stress pathways. The persistence of *CFTR* transcripts in CF secretory cells indicates that this cell type could be an attractive therapeutic target for *CFTR* gene editing and/or transfer in the more accessible airway surface epithelium of the trachea. Further efforts are required to fully understand CFTR function in specific cell types (i.e., ionocytes vs secretory cells) in the different regions of the lung.

### 4.3. The Role of Secretory Cells in the Small Airways

Airway secretory club cells have been thoroughly studied for their capacity to secrete host innate defense proteins (e.g., CCSP, LTF, and SLPI) and metabolize inhaled xenobiotics (e.g., p-xylene, benzo(a)pyrene, and ethylnitrosourea) that can be cytotoxic and/or carcinogenic [87,88,89,90]. One of the more prominent characteristics of this cell type is the plasticity of these cells that act as progenitor cells for specialized airway epithelial cells, e.g., ciliated and goblet cells. As suggested by recent transcriptomic data, secretory club cells express ion channel genes, including *CFTR* and *SCNN1*, and, therefore, maintain the balance of salt and water in the small airways [85]. However, only a few studies have characterized ion transport activity in the small airways and the potential roles for secretory club cells in controlling ion and water movements across the epithelium in the distal lung [91,92,93,94]. Blouquit et al. demonstrated that disease-controlled small airways exhibit active ENaC-mediated Na^+^ absorption and CFTR-mediated Cl^−^ secretion, while CF small airways failed to modulate ion fluxes and maintain ASL homeostasis [93]. Van Scott et al. purified rabbit club cells to confirm the presence of basal Na^+^ absorption and inducible Cl^−^ secretion in this cell type [95]. More recently, Kulaksiz et al. confirmed expression, localization, and function of CFTR in club cells of rat and human small airways [96]. Since both MUC5B and CFTR localize to the secretory club cells, and since secretory club cells provide host defense mechanisms, this cell type should be considered as a multi-dimensional mucin/ion/host defense regulatory cell that plays a crucial role in regulating mucosal defense and airway clearance, particularly in the small airways. Cell type-specific mechanistic studies will be important to determine how a single cell type regulates multiple secretory functions.

## 5. CFTR Malfunction Affects Mucus Properties

Ionic fluxes across the airway epithelium control the movement of water and electrolytes in the lungs. Hence, changes in the rate of ion diffusion affect ASL homeostasis and initiate a cascade of events leading to airway obstruction, inflammation, and bacterial infection. In CF, defective CFTR introduces a range of physiological alterations in both the airways and the SMGs, which alter mucin biochemical interactions and, subsequently, mucus viscoelasticity. Changes in Cl^−^, HCO_3_^−^, and Na^+^ movements influence airway hydration, chelation, ASL pH, and oxidative stress [2,3,4,97]. Whether occurring individually or in combination, these key biological processes govern the density and nature of inter- and intra-molecular bonds organizing the mucin network and are described in more detail below.

### 5.1. Effects of Inflammation on Mucus

Inflammation plays a critical role in the progression of CF lung disease and can affect the mucin network. Typically, in the lungs, bacterial colonization precedes inflammation, but in young CF patients, inflammation has been detected before the onset of infection, suggesting that aberrant mucus alone triggers an immune reaction in CF [19]. The recruitment of neutrophils to mucus-obstructed airways can worsen the viscoelasticity of mucus by distinct mechanisms. Neutrophils entrapped in static mucus ultimately undergo cell death, which is accompanied by the release of large nuclear DNA molecules that interweave with mucin polymers, adding complexity to the network [98,99]. The remarkable efficacy and widespread use of inhaled rhDNase for the treatment of CF patients validated the notion that breaking down large molecules contributing to mucus plugs improved airway clearance [100,101]. Alongside extracellular DNA, polymorphonuclear leukocytes release active enzymes, such as neutrophil elastase (NE) and myeloperoxidase (MPO), both affecting the mucin polymeric network [102]. NE indirectly upregulates ENaC function by degrading a host defense protein, short-palate *lung,* and nasal epithelial clone 1 (SPLUNC1), and thus exacerbates water hyperabsorption in inflamed CF airways [103]. MPO catalyzes the oxidation of free thiols by the reaction of oxygen peroxide with thiocyanate and therefore intensifies disulfide bond formation between cysteine residues distributed throughout the mucin protein backbone, causing mucus to stiffen [104]. Furthermore, lower glutathione concentrations, as detected in CF bronchoalveolar lavages, may worsen oxidative damage and increase mucus crosslinking [97,104]. Hence, inflammation can affect mucus biophysical properties in a myriad of ways and is set to start at a very young age for people living with CF. Despite inflammation playing an important role, CF muco-pathogenesis is primarily associated with the physiological consequences of CFTR malfunction, in particular reduced Cl^−^, HCO_3_^−^ secretion, and increased Na^+^ absorption.

### 5.2. Low HCO_3_^−^ and pH Influence Mucin Network Organization

Impaired CFTR-mediated HCO_3_^−^ secretion affects both polycation chelation and ASL pH. Inside goblet cells, large and heavily glycosylated mucin proteins are tightly packed within small ~1 µm granules as a result of a high [Ca^2+^] and [H^+^] environment that ensures nematic arrangement and polyionic charge shielding [105]. Upon secretion, electrolyte composition and concentrations are adjusted to the physiological levels of the extracellular milieu. In the ASL of healthy individuals, high HCO_3_^−^ concentration elicits a chelation process that is critical for proper mucin expansion via the exchange of one Ca^2+^ for two Na^+^ ions [106]. In CF, chronic HCO_3_^−^ deficiency produces partial Ca^2+^ sequestration, causing incomplete unfolding of the mucins [4,107]. Maintenance of secreted mucins in a semi-expanded state can contribute to inadequate mucus transport and/or difficulties slithering through narrow tubular structures (e.g., small airways, SMG ducts, pancreatic ducts, and vas deferens), which are all affected in CF. Furthermore, the acidic pH inside the mucin granules generates non-covalent hydrogen bonds to prevent electrostatic repulsion from the negatively charged terminal sugars. Once released in the more alkaline ASL environment, hydrogen bonds are disrupted, allowing for the mucin network to relax and swell via the Donnan effect [106]. In a low-pH environment, such as gastric pH (pH 2), side chains are maintained in a protonated state, which compromises the relaxation of the polymeric network, increases gel viscoelasticity, and slows mucus dynamics [108,109].

In CF nasal airways and in in vitro CF models, ASL pH was measured at values around 6.5, which is considered as mild acidification and should allow for the deprotonation of most carboxyl side chains [110,111]. However, in non-CF cultures exposed to high CO_2_ concentrations (to lower pH), ASL viscosity increased. Under the same high CO_2_ conditions, adding NaCl but neither iodoacetimide (to block disulfide bond formation) nor HCO_3_^−^ (to promote chelation without affecting hydrogen bonds) normalized ASL viscosity, suggesting that electrostatic interactions play an important role in the aberrant CF mucus properties [21]. Remarkably, alterations of mucus viscoelasticity by lowering pH were less significant (2.5-fold increase) than the effects of hyperconcentration (logarithmic scale increase) on both airway and gastrointestinal mucus [108,109,112].

### 5.3. Mucin Hyperconcentration Alters Mucus Viscous and Elastic Behavior

Defective CFTR reduces Cl^−^ secretion and upregulates ENaC-mediated Na^+^ absorption, leading to subsequent water hyperabsorption across the epithelium and subsequent ASL dehydration or lack of hydration [3,113]. Moreover, goblet cell and submucosal gland hyperplasia are responsible for increased mucus production in CF airways. The combination of mucus overproduction and lack of hydration increases the solidity of airway secretions. Hence, measurement of mucus concentration is paramount to the study CF lung disease and investigation of novel therapeutics aimed at restoring airway clearance in CF.

CF sputum concentration was measured at 8–10% solids, which is fivefold above normal sputum values [114]. Concentration-dependent polymeric gel behavior is governed by the distance between the monomers or, in the case of mucus, the mucin dimers [32,115]. In polymer physics, as water content decreases, mucin concentration (*c_m_*) transitions from an overlap concentration (*c** at ~1 mg/mL) to an entanglement concentration (*c^e^* > 25 mg/mL for pure mucins), which affects the rheological properties of the mucus gel by orders of magnitude. In the semi-diluted untangled regime (*c* < c_m_* < *c^e^*), mucin chains overlap and begin to interact depending on the mucin size, as defined by the molecular weight and radius of gyration. In an entangled regime (*c_m_* > *c^e^*), mucin polymer chains interpenetrate and form additional covalent and non-covalent bonds, stiffening the network and increasing reptation time. For concentrated mucus (>3% solids), imposed topological constraints affect the relaxation time and diffusion of the mucin chains, hindering motion and the overall dynamics of the network. In CF, airway mucus concentration often exceeds 5% solids and, therefore, exhibits both viscous and elastic behavior with longer time associated with large scale motion (i.e., slower dynamics) [114].

Increasing mucin chain proximity, as a consequence of mucus hyperconcentration or lack of hydration, not only restricts polymer chain motion, but also creates additional protein–protein interactions [109]. In healthy mucus (~2% solids), electrostatic repulsion due to the negative charges decorating the oligosaccharide side chains controls mucus swelling and polymeric network configuration. The distance between amino acid residues with ionizable side chains is critical to generate intermolecular salt bridges and requires less than 4 Å of physical distance. Once formed, the stability of salt bridges depends on pK*_a_*, supporting the importance of pH in mucus biochemical interactions. Overall, CFTR dysfunction increases the number of inter- and intra-molecular bonds, further exacerbating the complex chemical landscape of CF mucus.

## 6. Biochemical Alterations Impair Mucus Transport

Altered mucin interactions due to entanglement, compaction, and/or excess chemical bonding translate into aberrant mucus viscoelasticity, which affects transport by the cilia and passage through the gland ducts. In clinical studies, pulmonary function (e.g., FEV_1_) negatively correlated with the elastic and viscous moduli of CF sputum, and a similar relationship was shown between airway clearance and sputum % solids [114]. In in vitro models, increased mucus production is associated with an upregulation of several genes involved in ion transport, including *CFTR*, *TMEM16A*, and multiple members of the solute carrier (*SLC*) family, and a downregulation of *SCNN1* (ENaC) genes, suggesting that mucin release and ionic fluxes require synergistic regulation to ensure mucus flow [65]. Hence, failure to upregulate CFTR expression in response to mucin secretion will inevitably affect the mucus gel properties.

### 6.1. In Vitro CF Models Replicate Impaired Mucociliary Transport

As described earlier, primary HBE cell models harvested from non-CF and CF lungs have been studied for decades to understand the pathogenesis of CF and, more recently, to test CFTR modulator therapies prior to clinical testing. Once transitioned to an air–liquid interface, HBE cell cultures differentiate into a pseudostratified epithelium within four weeks and reproduce basic cellular functions important for lung defense mechanisms, more precisely, mucus secretion and ciliary transport [116,117]. Similar to the airway surface epithelium, HBE cell cultures control ASL height and composition via tight regulation of ion channels. To examine slight changes in ion fluxes, transepithelial electrical measurements can be performed via Ussing chambers. This powerful assay was crucial in the discovery of the first CFTR modulator in high-throughput screening and later complemented with intestinal organoid swelling assays [9,118]. Despite the lack of SMGs, non-CF HBE cell cultures maintain constant mucus movement under thin-film conditions, while CF HBE cell cultures are unable to transport mucus under the same conditions, which correlates with a decrease in PCL height and pH, as well as an increase in mucin concentration [113,119]. HBE cell cultures were utilized to show that mucus hyperconcentration, which is defined by concentrations >2% solids, causes an increase in the osmotic pressure of the mucus layer that compresses the cilia and hinders transport [32]. In a Calu3 cell line deficient for CFTR expression, lack of channel function caused increased secretion and concentration of both MUC5AC and MUC5B proteins, while no increase in mucin gene expression was detected, suggesting that the abnormal properties of CF mucus stimulate the release of mucin granules independently of *MUC* gene upregulation [119]. One limitation of in vitro models is the absence of neutrophil infiltration, a hallmark of CF mucus plugs that further exacerbates the viscoelastic properties of mucus due to extracellular DNA entanglement and/or ROS crosslinking, as described earlier. Nevertheless, HBE cell cultures have been pivotal in the advancement of our understanding of CF and the development of novel CF therapies.

### 6.2. Defective Functioning of the CF Submucosal Glands

SMGs are essential sac-like structures distributed throughout the large airways that act as a mucus reservoir to ensure copious and fast release of mucus into the proximal airways in response to a wide range of stimuli (e.g., inhaled pathogens, particles, and irritants). Adrenergic and cholinergic signaling stimulate the secretion of two cell types populating the gland acini, the serous cells that secrete a low viscosity fluid and the mucous cells that secrete a high viscosity fluid [27]. Prior to release into the airway lumen, the mixture of serous and mucous secretions passes through a narrow structure composed of the collecting and the ciliated ducts, which forces the SMG mucus to adopt a strand-like configuration that subsequently is transported toward the glottis by the beating of the cilia while sweeping the airways to clear particles [120,121,122]. CF animal models that possess SMG (e.g., pigs, sheep, ferrets, and rats) revealed that the functioning of the SMG was compromised as early as birth or shortly after the development of the glands [120,123,124,125]. Live microscopy techniques showed that mucus emerging from the SMG remained attached to the airway surfaces, which affected MCT in vivo and was associated with impaired antimicrobial properties [120,122]. A recent study showed that, similar to surface mucus, CF gland mucus present a solids content higher than non-CF gland mucus (6% vs 4% solids, respectively), which correlated with increased osmotic pressure and the collapse of cilia in the ciliated duct [31]. In contrast with mucus produced by the surface epithelium, SMG mucus failed to dissolve in excess PBS, as shown in the same study. Insoluble properties of SMG mucus were consistent with reduced sulfation of O-glycans shielding the MUC5B-rich glandular mucus. Additionally, glandular mucus colocalized with a proline-rich protein, PRR4, that may act as a mucin cross-linker. In diseases like non-CF bronchiectasis (NCFB) and primary ciliary dyskinesia (PCD), the sputum concentration of PRR4 was significantly elevated compared to disease controls, which is consistent with SMG hypersecretion. However, PRR4 concentration was significantly reduced in CF sputum, which correlated with low PRR4 staining in CF bronchoalveolar lavages, consistent with SMG hyposecretion. This study concluded that the cohesion forces applied by CF SMG mucus are disproportionate with the size of the ducts and, as a result, MUC5B-rich mucus is retained inside the gland and acts as a cork, as shown in Figure 2. These findings are consistent with observations of CF pigs and explain reduced antimicrobial factor secretion in CF lungs [120,122,126].

### 6.3. CFTR Rescue Reverses Mucus Abnormality In Vitro

In 2019, ETI/Trikafta (the CFTR modulator combination) became available to eligible patients and to the broad scientific community, providing an invaluable tool to study the biochemical and biophysical properties of mucus in cells originating from the same patient before and after CFTR rescue. A recent study examined changes in mucus properties in CF HBE cells obtained from patients homozygous for F508Del CFTR in response to ETI treatment via various assays to determine how CFTR modulators affect the mucin network organization [119]. Bioelectrical measurements confirmed that a 3-day treatment restored CFTR function but failed to normalize the mild acidification detected in CF HBE cell cultures. However, the same treatment regimen significantly decreased mucin concentration, as well as reducing the total amount of mucins by ~30%, contrasting with CFTR-KO Calu3 cells that exhibited increased mucin concentration and secretion. The same study showed that ETI treatment facilitated mucus removal from the cell surfaces via short (15 min) cell washings (70% of total mucus removed), while ~threefold more mucus remained attached to the cell surfaces of untreated CF cells (see Figure 3). Wash solutions adjusted for pH or HCO_3_^−^ concentration failed to remove the adherent mucus from untreated CF cells. In contrast, long (1 h) washings removed ~70% of total mucus from untreated CF cells, suggesting that extended hydration reverses the aberrant properties of CF mucus. Biophysical analyses of cell washings revealed a similar profile of homogenous and thick viscoelastic fluid for long washes and ETI-treated cells, while short cell washes depicted a distinct rheological profile of heterogeneous and watery material (i.e., fluid containing compact mucus flakes), indicating that CF mucus requires time for entanglement to subside and non-covalent bonds to break. This study demonstrates that mucus hydration is the dominant biochemical adjustment of CF mucus in response to CFTR modulator treatment. In vivo, ETI works systemically and, although not yet demonstrated, is expected to have similar effects on SMG gland mucus. Clinical data support this notion, as patient treated with ETI showed significant improvement in airway clearance and increased secretion of antimicrobial factors [17,127]. Granting that CFTR function in SMGs is essential for proper airway clearance, gene editing therapies aimed at restoring CFTR expression/function in acinar serous and/or mucous cells will pause additional challenges compared to surface epithelia due to intricate access.

## 7. Discussion and Future Directions

Mucus abnormalities play a central role in CF pathogenesis and have long been the target of therapeutic approaches to treat the symptoms of CF. Unlike previous therapies, CFTR modulators address the underlying cause of CF and reverse mucus abnormalities by restoring CFTR function in a systemic manner. With both ivacaftor and ETI available for children and adults, the vast majority of individuals with CF can now benefit from highly effective modulator therapy. However, there is still a demand for more comprehensive therapeutic strategies, including CFTR gene transfer/editing and cell-targeted therapies, to improve outcomes for individuals with CF who have ultra-rare CFTR mutations or develop adverse effects to the highly effective modulator therapies. With the ultimate goal of a path to a cure for CF, it is important to advance our understanding of cell type and region-specific roles for CFTR and secretory mucin regulation.

## Figures and Tables

**Figure 1 ijms-23-10232-f001:**
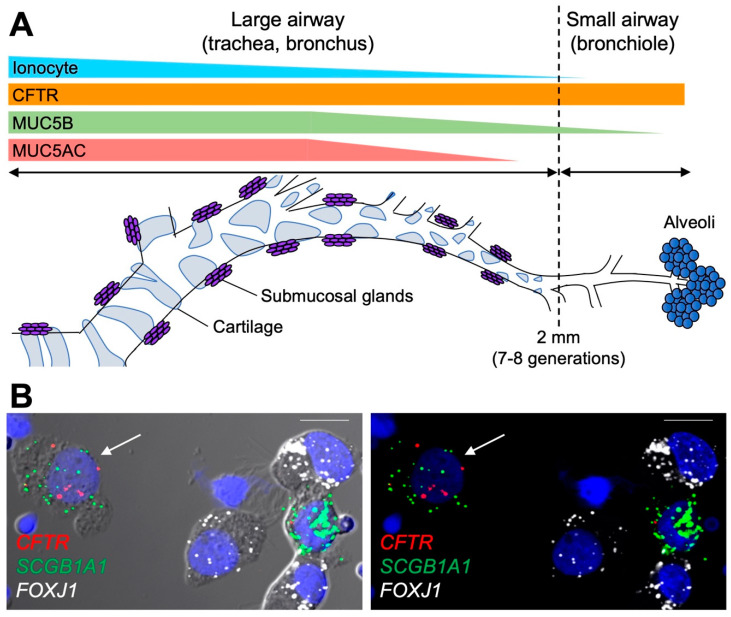
Intraregional expression of secretory mucins and CFTR in human airway superficial epithelia. (**A**) Regional distribution of secretory mucins, ionocytes, and *CFTR* transcripts in human conducting airway superficial epithelia. In the proximal (large) airways, both MUC5B and MUC5AC are expressed by the superficial epithelium. In the distal (small) airways, MUC5AC expression ceases, while MUC5B expression persists. *CFTR* is expressed by the superficial epithelium throughout the entire conducting airways, whereas the frequency of ionocytes is reduced in the distal airways. (**B**) RNA in situ hybridization (RNA-ISH) on isolated small airway epithelial cells. Cells were isolated from freshly excised small airway tissue obtained from a non-diseased transplant donor lung. Using RNA-ISH probes, *CFTR* colocalized with secretory cell marker transcripts (*SCGB1A1*) in a non-ciliated cell (arrow) and not with *FOXJ1*-positive ciliated cells. Left image shows differential interference contrast and fluorescent image overlay. Right image shows fluorescent image only. Scale bar = 10 μm.

**Figure 2 ijms-23-10232-f002:**
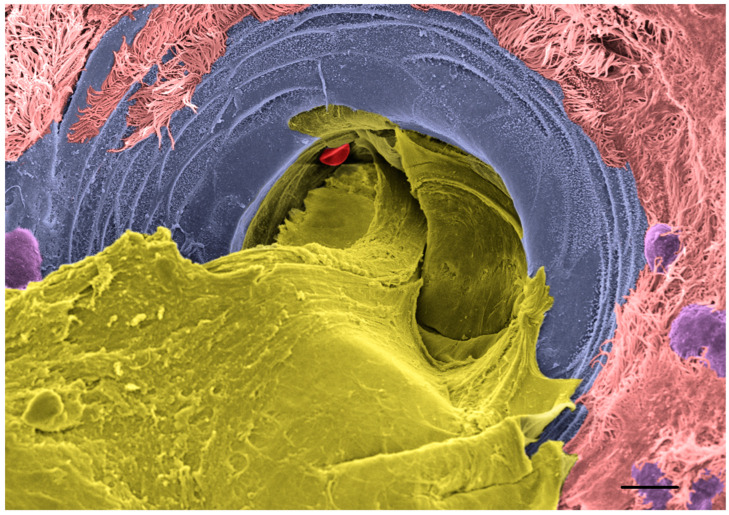
Scanning electron microscope (SEM) image of a CF submucosal gland duct obstructed by mucus. Large airways were collected from a CF lung at the time of transplant, stimulated with methacholine (10 µM) for 30 min and processed for SEM. Image was acquired with a Zeiss Supra 25 SEM and colorized using Adobe Photoshop software. Color schemes show mucus in yellow, cilia in pink, gland duct in blue, secretory cells in purple, and a red blood cell in red. Scale bar 10 μm.

**Figure 3 ijms-23-10232-f003:**
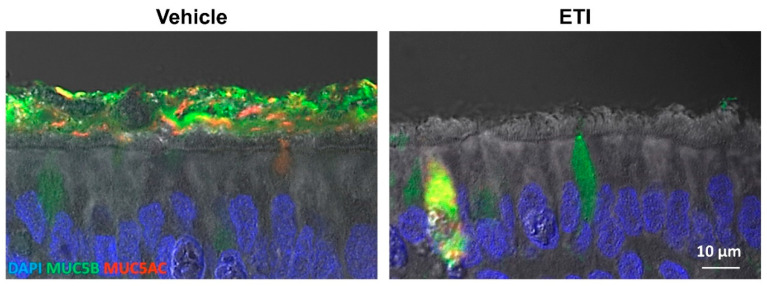
Effects of ETI treatment on mucus removal from the cell surfaces. Primary HBE cells homozygous for F508del were treated for 3 days with 0.06% dimethyl sulfoxide (DMSO/vehicle) or 3 µM VX-661, 2 µM VX-445, and 1 µM VX-770 (ETI). Cells were washed for 15 min with PBS, fixed with 4% PFA, and processed for immunohistochemistry. Representative images of histological sections stained with MUC5AC (red), MUC5B (green), DAPI (blue), and DIC overlay.

## Data Availability

Not applicable.

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
