# Peer review of "Mucins and CFTR: Their Close Relationship"

_ijms, 2022, doi:10.3390/ijms231810232_

Round 1

Reviewer 1 Report

This review by Kenichi Okuda et al focuses on the relationship between failed mucociliary clearance and a genetic disorder Cystic Fibrosis. The review is written well, the topic is relevant and addresses a specific gap in the field and also covers a vast literature related to the field. The review outlines the single cell transcriptomic analysis and cellular specificity of the mucin secreting cells in human airway superficial epithelia and highlights the disturbance of mucus homeostasis in the Cystic Fibrosis disease. Specifically, the close regional specificity between Cl- and HCO3- transporter CFTR (mutated in Cystic Fibrosis) and mucin producing cells highlights the role of ion transport and mucus homeostasis in disorders. The review emphasizes on the need to develop more effective therapeutic strategies for CFTR based on the genetic manipulation and the need to understand the regional specific cell targets to ensure successful gene therapy of the targeted epithelial cells in the lungs.

Author Response

Thank you for taking the time to read our review and for the kind feedback.

Reviewer 2 Report

This is a comprehensive, well-developed and well-written review on the relationship between CFTR function and the properties of airway mucins in the context of CF that should be of high interest to the field. I have only some minor stylistic suggestions that the authors may consider (or ignore).

Specific Points:

-          Pg1/last line: before delving into the history of CFTR modulator therapies, it may help the non-expert reader if you could shortly mention the different classes of CFTR mutations, or at least mention that CFTR mutations may affect gene expression (e.g. non-sense), protein processing/folding, and/or channel function/gating.

-          Pg3/Section2.2/second-to-last sentence “…suggesting a limited capacity for small airway MCC in diseases  [32].” Please rephrase for clarity (e.g. small airway MCC has limited capacity. Thus, impairments in small airway MCC are prone to cause disease…).

-          Pg3/Section2.3/last sentence: Please detail WHY the “…abundant and irregular MUC5B/MUC5AC mucous flakes…” are indicators of distal airway disease. Please note that the distribution of MUC5B/MUC5AC in the respiratory tree is only discussed later in the manuscript (Section 3.3).  

-          Pg.6/Ln11: Please rephrase “superficial epithelium of disease-control, …” for clarity (e.g. …in the normal or non-CF superficial epithelium…)

-          Pg6/Ln13: “…in CF trachea”; I would suggest to use “in the CF trachea” or “in CF tracheas”

-          Pg8/Ln4: please define “SPLUNC” at first mention

-          Pg9/second paragraph starting with “Increasing mucin chain proximity…”. Please revise for clarity. How does CF affect distance between amino acid residues? Does CF affect folding and/or “..stability of salt bridges once formed…”.

-          Pg11/Section 6.3.: to jog the readers memory, it could be helpful to replace “ETI” with “ETI/Trikafta” or define/describe the term again (e.g. the CFTR modulator combo ETI).

Reviewer 3 Report

Okuda et al.
The authors present a comprehensive review on the relationship between Mucins MUC5AC/MUC5B and CFTR. They discuss roles of CFTR in the lungs, the role of mucins and mucociliary clearance, single cell transcriptomics that have lead to new insights on pulmonary cells expressing CFTR and how CFTR dysfunction affects mucus properties and clearance.

I found the review to be up-to-date and comprehensive. The review presents a lot of pertinent information at one place and will be a ready guide to people who want to read about CFTR and mucociliary clearance. The review is very well written, and easy to read. I have no specific details to add or ask the authors.

Author Response

(The authors gave the same response as above.)
